# Short- and Long-Term Effect of Cochlear Implantation on Disabling Tinnitus in Single-Sided Deafness Patients: A Systematic Review

**DOI:** 10.3390/jcm11195664

**Published:** 2022-09-26

**Authors:** Samar A. Idriss, Pierre Reynard, Mathieu Marx, Albane Mainguy, Charles-Alexandre Joly, Eugen Constant Ionescu, Kelly K. S. Assouly, Hung Thai-Van

**Affiliations:** 1Department of Audiology and Otoneurological Evaluation, Edouard Herriot Hospital, Hospices Civils de Lyon, 69002 Lyon, France; 2Department of Otorhinolaryngology and Head and Neck Surgery, Eye and Ear University Hospital, Holy Spirit University of Kaslik, Beirut 1202, Lebanon; 3Institut de l’Audition, Institut Pasteur, University of Paris, INSERM, 75012 Paris, France; 4Faculty of Medicine, University Claude Bernard Lyon 1, 69100 Villeurbanne, France; 5Department of Otology, Otoneurology and Pediatric Otolaryngology, Pierre-Paul Riquet Hospital, Toulouse Purpan University Hospital, 31300 Toulouse, France; 6Brain and Cognition Laboratory, UMR 5549, Toulouse III University, 31062 Toulouse, France; 7National Commission for the Evaluation of Medical Devices and Health Technologies, Haute Autorité de Santé, 93210 La Plaine St Denis, France; 8Department of Otorhinolaryngology and Head & Neck Surgery, University Medical Center Utrecht, 3584 CX Utrecht, The Netherlands; 9UMC Utrecht Brain Center, University Medical Center Utrecht, 3584 CX Utrecht, The Netherlands; 10Cochlear Technology Centre, 2800 Mechelen, Belgium

**Keywords:** single-sided deafness, cochlear implant, disabling tinnitus, systematic review, speech perception, sound localization, hyperacusis, quality of life

## Abstract

Patients with single-sided deafness can experience an ipsilateral disabling tinnitus that has a major impact on individuals’ social communication and quality of life. Cochlear implants appear to be superior to conventional treatments to alleviate tinnitus in single-sided deafness. We conducted a systematic review to evaluate the effectiveness of cochlear implants in single-sided deafness with disabling tinnitus when conventional treatments fail to alleviate tinnitus (PROSPERO ID: CRD42022353292). All published studies in PubMed/MEDLINE and SCOPUS databases until December 2021 were included. A total of 474 records were retrieved, 31 studies were included and were divided into two categories according to whether tinnitus was assessed as a primary complaint or not. In all studies, cochlear implantation, evaluated using subjective validated tools, succeeded in reducing tinnitus significantly. Objective evaluation tools were less likely to be used but showed similar results. A short-(3 months) and long-(up to 72 months) term tinnitus suppression was reported. When the cochlear implant is disactivated, complete residual tinnitus inhibition was reported to persist up to 24 h. The results followed a similar pattern in studies where tinnitus was assesed as a primary complaint or not. In conclusion, the present review confirmed the effectiveness of cochlear implantation in sustainably reducing disabling tinnitus in single-sided deafness patients.

## 1. Introduction

Single-sided deafness (SSD), also known as unilateral profound hearing loss [1], is associated with a hearing impairment with higher perception of hearing handicap and visual annalog scores [2]. Despite normal or near-normal contralateral hearing status, monaural stimulation can lead to a wide range of audiological disabilities such as poor speech perception in noise and sound localization [3,4]. In addition, patients with SSD can experience an ipsilateral severe tinnitus [5,6,7]. These issues can have a crucial impact on individuals’ social communication and interaction, in addition to significant effects on their quality of life (QoL) [8]; it can also lead to a psychological distress [9].

Tinnitus severity is graded using various validated subjective tools such as Tinnitus Questionnaire (TQ) [10,11], Tinnitus Handicap Index (THI) [12], Tinnitus Reaction Questionnaire (TRQ) [13], Visual Analog Score (VAS) [14], Tinnitus Rating Score (TRS) [15], Subjective Tinnitus Severity Scale (STSS) [16], and Numeric Rating Scale (NRS) [17], among others. Severe disabling tinnitus is defined by a TFI > 32/100, THI > 58/100, TQ > 42/84, or VAS loudness or annoyance >6/10 [18]. It is a difficult-to-treat disabling condition, and is frequently associated with by hearing loss [19]. One of its main pathophysiological mechanisms involves a paradoxical enhanced central activity associated with loss of peripheral input [20]. Persistent bothersome tinnitus can be very harmful to psychological health [9,21] and co-occurs with several comorbidities [22]. Notably, it can be associated with sleeping disturbances, cardiovascular diseases, and metabolic disorders [23]. The American Academy of Otolaryngology and the European societies have published guidelines for the management of tinnitus [24,25]. Drugs, including antidepressants [26] anticonvulsants [27], and dietary supplements [28,29], as well as electromagnetic [30] or laser [31] stimulation, and acupuncture [32] are not recommended [24,25]. Psychological therapies such as cognitive behavioral therapy (CBT) [33,34] are recommended [24,25]. Tinnitus retraining therapy (TRT) [35], psychotherapy [36], relaxation and meditation [37,38], hypnosis [39], biofeedback [40], education-information [41], and stress management-problem solving [42], among others, can be helpful and reduce tinnitus [24,25]. In the absence of hearing loss, sound therapy, delivered via ear/headphones, may be recommended for bothersome tinnitus [25,43], and in the presence of hearing loss, hearing aids (HAs) are recommended [24,25]. In cases of severe hearing loss, cochlear implant (CI) appears to be superior to conventional treatments, including HAs, contralateral routing of sound HAs (CROS), and bone conduction hearing devices [44,45,46]. Consequently, CI was approved by the US Food and Drug Administration for SSD [47] and was recently considered as an indication for disabling tinnitus with SSD in France after insufficient effectiveness of conventional treatments [48].

To date, a number of studies have evaluated the effect of cochlear implantation in the treatment of disabling tinnitus in SSDs; however, only a few reviews are available [49,50]. In the first review, no studies with objective tinnitus assessment tools were included and the maximum follow-up period was up to 28 months [49]. In the second review, tinnitus assessment tools were also subjective and were limited to those using THI and/or VAS [50]. The present systematic review included all studies, published through December 2021, in which tinnitus was evaluated as a primary or non-primary complaint. Assessing tinnitus as a primary complaint reduces the risk of false-positive and false-negative errors [51]. Studies using subjective assessment methods, as well as those using objective assessment methods, were included. When it came to subjective methods, all validated questionnaires and scales were considered without any restrictions. Furthermore, the effect of cochlear implantation on tinnitus was not only analyzed in the short term, but also the long term.

The present systematic review aims to provide a comprehensive overview of the short- and long-term effects of cochlear implantation on disabling tinnitus in adults with single-sided deafness.

## 2. Materials and Methods

The review protocol is available on International prospective register of systematic reviews (PROSPERO) (ID: CRD42022353292). The Preferred Reporting Items for Systematic Reviews and Meta-Analysis (PRISMA) statement was used for this systematic review [52].

### 2.1. Search Strategy

A systematic search of published studies was performed in PubMed/MEDLINE and SCOPUS databases using the syntax (tinnitus [Title/Abstract]) AND single-sided deafness [Title/Abstract] AND Cochlear implant) and the different combinations (tinnitus AND single-sided deafness), (tinnitus AND cochlear implant), and (cochlear implant AND single-sided deafness). The search was conducted in December 2021. All published studies available at this time were included in the review process. The search terms included combined expressions and synonyms of tinnitus, single-sided deafness, and CI. These include ear ringing, buzzing, unilateral hearing loss, and intracochlear electrical stimulation.

### 2.2. Study Selection

All studies on cochlear implantation in adult patients with SSD and disabling tinnitus, in which tinnitus was evaluated as a primary or non-primary complaint, were selected. Studies where tinnitus was evaluated pre- and post- operatively, using subjective and/or objective tools, in the short- or long-term, were eligible. During screening, duplicates, systematic reviews, and articles written in languages other than English were excluded. Case reports and studies with overlapping study population were not excluded. Lack of previous therapeutic trials was not an exclusion criterion. Two reviewers, S.A.I. and P.R., screened each study (title/abstract) independently. Disagreements were resolved by a third reviewer. Studies were divided into two groups according to whether the primary complaint was tinnitus.

### 2.3. Quality Assessment

Two authors, S.A.I and K.K.S.A., independently assessed the risk of bias (RoB). We used the ROBINS-I tool (Risk Of Bias In Non-randomized Studies—of Interventions) to evaluate risk of bias [53]. The tool consists of seven domains: confounding, selection of participants, classification of interventions, deviation from intended intervention, missing data, measurement of outcomes, and selection of reported results. The criteria were defined and adapted to our research question about cochlear implantation for SSD with disabling tinnitus. Items were scored as low risk of bias, moderate risk of bias, serious risk of bias, or unclear based on the guidelines of the ROBINS-I tool. Consensus was obtained after discussion between the two reviewers.

### 2.4. Data Extraction

All study characteristics and outcomes were extracted by S.A.I. and P.R. independently. The primary outcome was the difference between pre- and post-operative evaluation of tinnitus on validated multi-item tinnitus distress questionnaires and/or objective evaluation measurements. Additional outcomes were also extracted including hyperacusis, sound hypersensitivity, speech perception, sound localisation, word recognition, quality of life, work performance, and psychosocial comorbidities.

## 3. Results

### 3.1. Search Strategy and Study Selection

A total of 474 records were retrieved, and 31 studies were included in the systematic review (Figure 1). Post-implantation tinnitus suppression was analysed in 479 patients using various assessment methods. These studies were divided into two groups; 14 studies in which the primary complaint was tinnitus, and 17 studies in which tinnitus was not the primary complaint. Some studies had an overlap in their population samples [54,55,56,57].

### 3.2. Quality Assessment of Included Studies

The critical appraisal can be found in Table 1 and Table 2 for studies where tinnitus was the primary complaint and those where tinnitus was not the primary complaint, respectively.

In studies where tinnitus was the primary complaint, only one study [58] defined appropriately its inclusion criteria. The remaining studies either did not provide information on contralateral ear [59] or included moderate hearing loss thresholds for inclusion criteria [54,55,60,61,62]. In addition, in several studies, the efficacy of conventional treatments was not tested before CI [56,63,64,65,66,67,68]. When selecting participants, inclusion and exclusion criteria were not well defined [55]. Two out of fourteen studies were retrospective [55,64]. Blinding was applied in only one study [62]. The population samples of two studies overlapped and the criteria for recruiting additional participants were not well defined [54,56]. The process of cochlear implantation and rehabilitation was not clear in all studies [60,64,67]. The intervention protocol was either unreported [55,58,59,63,64] or did not respect standard process [56,59,61,62,65,67]. Missing data, participant dropouts and withdrawal exceeding 10% [54,58,65] were justified in only one study [58].

**Table 1 jcm-11-05664-t001:** Quality assessment of studies in which tinnitus was the primary complaint.

ROBINS-I tool			Risk of Bias (RoB)						
Study	Study Design	Sample Size	Bias Due to Confounding	Bias in Selection of Participants	Bias in Classification of Interventions	Deviation from Intended Intervention	Bias Due to Missing Data	Bias in Measurement of Outcomes	Bias in Selection of Reported Result
Ahmed et al. [63]	PCS	13	●	O	O	Ø	Ø	◐	O
Arts et al. [62]	PCS	10	●	O	O	●	O	O	O
Holder et al. [64]	PCS	12	●	●	Ø	Ø	Ø	◐	O
Kleinjung et al. [60]	CR	1	●	NA	Ø	O	Ø	NA	Ø
Macias et al. [66]	PCS	16	●	O	O	O	O	◐	O
Mertens et al. [55]	RCT	23	●	●	●	Ø	Ø	◐	O
Mertens et al. [56]	PCS	11	●	O	O	●	Ø	◐	O
Poncet-Wallet et al. [65]	PCS	26	●	O	O	●	●	◐	◐
Punte et al. [59]	PCS	26	Ø	O	O	Ø	O	◐	●
Punte et al. [68]	PCS	7	●	O	O	●	O	◐	O
Ramos et al. [61]	PCS	6	●	O	O	●	O	◐	O
Song et al. [58]	PCS	9	O	O	O	Ø	●	◐	●
Van de Heyning et al. [54]	PCS	22	●	O	O	O	●	◐	◐
Zeng et al. [67]	CR	1	●	NA	Ø	●	Ø	NA	Ø

PCS: prospective cohort study; RCS: retrospective cohort study, CR: Case report. Confounding: O = no confounding (use of three inclusion criteria: SSD defined with (PTA (0.5, 1, 2, 4 kHz) > 70 dBs in one ear and <30 dBs in the other ear, severe tinnitus defined by TFI > 32, THI > 58, TQ > 42, VAS loudness or annoyance > 6/10, and failure of conventional treatment such as CROS, BCD, HA), ● = inclusion criteria not appropriately used, Ø = no information. Selection of participants (based on participant characteristics observed after the start of the intervention): O = no bias in selection of participants, ● = bias in selection of participants, NA: not applicable. Classification of interventions: O = intervention status well defined before application (CI), ● = intervention status defined retrospectively, Ø = no information. Deviation from intended intervention: O = standard cochlear implantation, activation and rehabilitation defined clearly in the protocol, ● = deviations to the intervention protocol, Ø = no information. Missing data: O = < 10% missing data, ● = ≥ 10% missing data, Ø = no information. Measurement of outcomes: O = similar measurement of outcomes between intervention groups AND blinding of the outcome assessors for intervention received by study participants, ◐ = similar measurement of outcomes between intervention groups AND no blinding of the outcome assessors for intervention received by study participants, ● = difference of measurement between groups AND no blinding of the outcome assessors for intervention received by study participants, NA: not applicable. Selection of reported results: O = primary outcomes reported according to the protocol, ◐ = primary outcomes reported for all groups (no subset) and explanation if missing data, ● = missing outcomes/data reported for a subset of measures, Ø: no information.

In studies where tinnitus was not the primary complaint, five studies defined appropriately its inclusion criteria [44,69,70,71,72]. When selecting participants, inclusion and exclusion criteria were not clearly provided [73,74], and blinding was not applied. Several studies were retrospective [71,72,74,75,76]. The CI intervention was not constantly described [57,77], and the majority of studies did not clarify if the standard CI protocol was adopted [44,69,71,72,74,75,76,78,79,80,81]. Missing data, participant dropouts and withdrawals exceeding 10% [71,74,75,76] were not constantly justified [75].

**Table 2 jcm-11-05664-t002:** Quality assessment of studies in which tinnitus was not the primary complaint.

ROBINS-I Tool			Risk of Bias (RoB)						
Study	Study Design	Sample Size	Bias Due to Confounding	Bias in Selection of Participants	Bias in Classification of Interventions	Deviation from Intended Intervention	Bias Due to Missing Data	Bias in Measurement of Outcomes	Bias in Selection of Reported Result
Arndt et al. [44]	PCS	11	O	O	O	Ø	O	◐	O
Buechner et al. [73]	PCS	5	●	●	O	O	O	◐	O
Dillon et al. [8]	PCS	20	●	O	O	O	Ø	◐	O
Dorbeau et al. [82]	PCS	18	●	O	O	O	O	◐	O
Finke et al. [75]	RCS	14	●	O	●	Ø	●	◐	Ø
Friedman et al. [71]	RCS	16	O	O	●	Ø	●	◐	◐
Gartrell et al. [77]	CR	1	●	NA	Ø	Ø	Ø	NA	Ø
Harkonen et al. [80]	PCS	7	●	O	O	Ø	O	◐	O
Haubler et al. [72]	PCS	20	O	O	●	Ø	O	◐	O
Kitoh et al. [81]	PCS	5	●	O	O	Ø	●	◐	O
Macias et al. [78]	PCS	16	●	O	O	Ø	O	◐	O
Mertens et al. [57]	PCS	15	●	O	Ø	●	O	◐	O
Peters et al. [83]	PCS	28	●	O	O	O	O	◐	◐
Sladen et al. [74]	RCS	23	●	●	●	Ø	●	◐	◐
Sullivan et al. [76]	RCS	60	●	O	●	Ø	●	◐	◐
Tavora-Vieira et al. [69]	PCS	9	O	O	O	O	O	◐	O
Tavora-Vieira et al. [70]	PCS	28	O	O	O	O	O	◐	O

PCS: prospective cohort study; RCS: retrospective cohort study, CR: Case report. Confounding: O = no confounding (use of three criteria: SSD defined with (PTA (0.5,1,2,4 kHz) > 70 dBs in one ear and <30 dBs in the other ear, and failure of conventional treatment such as CROS, BCD, HA), ● = inclusion criteria not appropriately used. Selection of participants (based on participant characteristics observed after the start of the intervention): O = no bias in selection of participants, ● = bias in selection of participants, NA: not applicable. Classification of interventions: O = intervention status well defined before application (CI), ● = intervention status defined retrospectively, Ø = no information. Deviation from intended intervention: O = standard cochlear implantation, activation and rehabilitation defined clearly in the protocol, ● = deviations to the intervention protocol, Ø = no information. Missing data: O = < 10% missing data, ● = ≥10% missing data, Ø = no information. Measurement of outcomes: O = similar measurement of outcomes between intervention groups AND blinding of the outcome assessors for intervention received by study participants, ◐ = similar measurement of outcomes between intervention groups AND no blinding of the outcome assessors for intervention received by study participants, ● = difference of measurement between groups AND no blinding of the outcome assessors for intervention received by study participants, NA: not applicable. Selection of reported results: O = primary outcomes reported according to the protocol, ◐ = primary outcomes reported for all groups (no subset) and explanation if missing data, ● = missing outcomes/data reported for a subset of measures, Ø: no information.

### 3.3. Data Extraction and Study Outcomes

#### 3.3.1. Tinnitus Evaluated as a Primary Complaint

In studies in which tinnitus was the primary complaint, pre- and post-operative tinnitus was evaluated, using numerous tools including validated questionnaires and scales, and objectives tests (Table 3). Validated self-reported instruments were used in all such studies, namely the Tinnitus Questionnaire (TQ) [10,11], THI [12], Tinnitus Reaction Questionnaire (TRQ) [13], VAS [14], Tinnitus Rating Score (TRS) [15], Subjective Tinnitus Severity Scale (STSS) [16], and/or Numeric Rating Scale (NRS) [17]. Objective measurements including electroencephalogram (EEG) along with functional imaging [58], and/or evoked and spontaneous cortical activities [67] were less frequently used. The follow-up period was variable studies and ranged between 12 min [67] to 36 months [55].

In all studies in which tinnitus was the primary complaint, early after implant activation, electrical stimulation succeeded to significantly reduce, sometimes completely, tinnitus loudness and distress [60,68]. VAS, THI, and TQ were used in the majority of studies, but also similar results were obtained with other tools such as TRQ, TRS, and STSS [63,65]. No tinnitus aggravation was noted in any of the included studies. Long-term (>12 months) tinnitus suppression was reported in several studies [54,55,65]. Tinnitus suppression was less likely to persist when CI was turned off [54,59,60,68]; persistence of suppression after CI deactivation was only reported in one study [61]. While some studies reported complete residual inhibition of tinnitus that ranged between a minute to 30 min [55,62,66], others reported that residual inhibition persisted for 12 [54] and 24 h [59,68]. Taken together, these results confirm the effectiveness of CI as a treatment in disabling tinnitus (Table 3).

Zeng et al. [67] assessed tinnitus presence objectively by recording cortical potentials and tinnitus loudness subjectively using a VAS. Evoked and spontaneous cortical activity was recorded in “tinnitus-presence” and “tinnitus-suppressed” conditions. Complete suppression of tinnitus was obtained after a low-rate low-level electrical intracochlear stimulation and was associated re-established brain activities. These results were coherent with a reduction of tinnitus loudness (VAS). In another study, Song et al. [58] explored EEG waves and activated Auditory Cortex (AC) areas by brain electromagnetic tomography among patients with tinnitus and SSD pre- and post-cochlear implantation; those with pre-operative enhanced activity in different regions of the AC, higher delta and gamma bands, and an increased connectivity between different area of the AC, were less likely to improve after CI. These results matched with NRS and TQ scores (Table 3).

#### 3.3.2. Tinnitus Evaluated as an Additional Complaint

In studies in which tinnitus was not the primary complaint, tinnitus was also investigated via validated questionnaires and scales including VAS, THI, TRQ, TQ, and/or tinnitus handicap questionnaire (THQ) (Table 4). No objective measurements were used.

All studies in which tinnitus was not the primary complaint reported tinnitus suppression. Among the 296 patients included in these studies, tinnitus was not suppressed in only one patient [75]. Tinnitus suppression remained stable over time [57,70,76,77]. When CI patients were compared to a control group, THI scores were significantly lower [83]. No objective measurements were applied for tinnitus in any of these studies (Table 4).

**Table 4 jcm-11-05664-t004:** Characteristics of studies investigating CI in SSD patients with disabling tinnitus in which tinnitus was not the primary complaint.

Study	Patients’ Criteria	n	Evaluation	Interval Studied	Results	Conclusion
Arndt et al. [44]	CI in SSD and tinnitus refractory to conventional treatment	11	*Tests* - HSM sentence test (speech comprehension in noise) - OLSA sentence test (speech comprehension in noise and speech localization) *Questionnaires* - SSQ - HUI3 - IOI-HA (QoL and outcome with hearing devices) - VAS (tinnitus)	6 months	- Significant improvement of speech localization and comprehension - Significant improvement of QoL - Significant reduction or complete suppression of tinnitus when present	- CI improved hearing abilities and was superior to the alternative treatment options - CI use did not interfere with speech understanding in the normal hearing ear
Buechner et al. [73]	CI in SSD and tinnitus	5	*Tests* - FST and HSM sentence test (speech comprehension in noise) - OLSA sentence test (speech perception and localization). *Questionnaires* - Sound quality - VAS (tinnitus)	12 months	- Significant benefit of speech perception tests (NB = 3) - None of the participants judged CI sound quality as intolerable - Significant suppression (NB = 3) or reduction (NB = 2) of tinnitus	- CI improved hearing and tinnitus
Dillon et al. [8]	CI in SSD and tinnitus	20	*Questionnaires* - Speech localization and perception - Traditional scores and SSQ subscales (QoL) - APHAB (difficulty) - THI (tinnitus)	12 months	- Improvements in speech perception in noise, spatial hearing, and listening effort - Significant improvement of QoL and less perceived difficulty - Significant reduction of tinnitus severity	- CI may offer significant improvement in QoL, reduction in perceived tinnitus, and subjective improvement in speech perception and hearing
Dorbeau et al. [82]	CI in SSD and tinnitus	18	*Tests* - Sound localization - SRT in quiet and noise (speech understanding in noise) *Questionnaires* - SSQ - GBI (QoL) - THI (tinnitus)	12 months	- Significant improvement of speech localization - No significant SRTs difference when speech and noise were co-located, but significantly better SRTs when speech and noise spatially separated. - Significant improvement of SSQ - Significant improvement of QoL - Significant reduction of tinnitus severity	- Strong significant and consistent CI benefits were observed for localization, speech performance, tinnitus reduction, and QoL
Finke et al. [75]	CI in SSD and tinnitus	14	*Tests* - FST and HSM sentence test in quiet and noise (speech perception and sound localization). *Questionnaires* - Sound localization - Fear to lose the second ear - QoL - Tinnitus and noise sensitivity	53 months	- Significant improvement of sound localization and sound quality - Substantial change in QoL - Reduction of tinnitus (*n* = 13); only one patient stated that the CI failed to reduce tinnitus	- Overall sense of increased well-being explained by the four different core categories localization, tinnitus, fear of hearing loss and QoL
Friedman et al. [71]	CI in SSD and tinnitus	16	*Tests* - Sound localization - CNC monosyllabic words and AzBio sentences (speech perception) - BKB-SIN or HINT (hearing in noise) *Subjective assessments* - Integration ability - Tinnitus	12 months	- Significant improvement in speech perception - No significant difference in sound localization - Improvement in integration ability - Suppression of tinnitus	- CI improved speech perception and performance, integration ability, and tinnitus
Gartrell et al. [77]	CI in SSD and severe tinnitus refractory to medical therapies	1	*Tests* - Sound localization - Speech in noise test - Audiometric threshold - HINT (hearing in noise) - CNC (speech discrimination) - IEEE sentence test (speech quality) *Questionnaires* - TRQ - TQ - THI	18 months	- Significant improvement of sound localisation - Improved speech intelligibility - Marked tinnitus reduction and remained over 16 months	- CI improved sound localization accuracy when compared and reduced tinnitus handicap
Härkönen et al. [80]	CI in SSD and tinnitus	7	*Tests* - Sound localization - Bisyllabic Finnish words (speech in noise test) *Questionnaires* - GBI (QoL) - SSQ and VAS (QoH) - Working performance and work-related stress - VAS (tinnitus)	28 months	- Significant positive effect of sound localization, speech perception in noise, QoL, and QoH - Improved working performance - Decreased tinnitus perception	- CI improved QoL, QoH, sound localization, speech perception in noise, work performance, and tinnitus
Häußler et al. [72]	CI in SSD and tinnitus refractory to conventional treatment	20	*Tests* - Speech perception - Hearing ability. *Questionnaires* - NCIQ (health related QoL) - SF-36 (general QoL) - Psychological comorbidities - TQ (tinnitus)	36 months	- Significant improvement of speech perception - Significant improvement of heath related QoL - Significant decrease of anxiety symptoms - Significant reduction of tinnitus	- CI improved hearing, tinnitus, QoL, and psychological comorbidities
Kitoh et al. [81]	CI in SSD patients	5	*Tests* - Sound localization - Japanese monosyllable test (speech perception in quiet and noise) *Questionnaires* - THI (tinnitus disturbance)	12 months	- Improvement of speech perception and increased sound localization accuracy - Reduction of tinnitus	- CI improved speech perception, sound localization, and tinnitus
Macias et al. [78]	CI in SSD and disabling tinnitus and hyperacusis refractory to conventional treatment	16	*Questionnaires* - HUI3 (QoL) - SSQ (hearing quality) - SHQ (hyperacusis) - THI and VAS (tinnitus)	12 months	- Substantial reduction in sound intolerance - Increase QoL - Substantial decrease of tinnitus	- CI improved tinnitus, hyperacusis, and QoL
Mertens et al. [57]	CI in SSD and disabling tinnitus	15	*Tests* - SRT in noise in non-tinnitus ear in CI-on and CI-off conditions *Questionnaires* - VAS and TQ (tinnitus)	36 months	- Significant improvement of speech perception and SRT - Improvement of TQ and remained stable or became better for 3 years - Significant decrease of VAS	- CI improved speech perception and tinnitus
Peters et al. [83]	CI and bone conduction devices in SSD and tinnitus	28	*Tests* - Sound localization - USTARR (speech recognition in noise) *Questionnaires* - SSQ - APHAB - GBI (QoL) - TQ and THI (tinnitus)	6 months	- CI had better speech reception, sound localization, TQ and THI - All treatment options had an improvement of disease specific QoL - Significant decrease of tinnitus	- CI group had better sound localization and perception, and decreased tinnitus burden
Sladen et al. [74]	CI in SSD and tinnitus	23	*Tests* - CNC word and AzBio sentence in quiet and noise (speech perception) *Other* - Tinnitus assessment tool	6 months	- Significant improvement of both word and sentence scores in quiet - No significant improvement of speech recognition in noise - Reduction in tinnitus severity	- CI improved speech understanding and reduced tinnitus
Sullivan et al. [76]	CI in SSD patients and tinnitus	60	*Tests* - Sound localization - CNC word and AzBio sentence in quiet and noise (speech perception) - Adaptive HINT (binaural hearing) *Questionnaires* - THQ (tinnitus)	72 months	- Sound localization tended to improve - Significant improvement of speech perception - Improvement of tinnitus; kept stable for many years	- CI meaningfully improved word understanding, tend to gradually improve sound localization, and improve tinnitus
Tavora-Vieira et al. [69]	CI in SSD and tinnitus	9	*Tests* - BKB sentence in noise (speech perception). *Questionnaires* - SSQ (hearing perception) - TRQ (tinnitus)	3 months	- Improvement of speech perception in noise - Significant improvement of hearing perception - Improvement of tinnitus	- CI improved speech understanding in noise, hearing perception, and tinnitus control
Tavora-Vieira et al. [70]	CI in SSD with tinnitus	28	*Tests* - BKB-SIN (speech perception) *Questionnaires* - SSQ (speech perception) - APHAB (hearing difficulties) - TRQ (tinnitus disturbance)	24 months	- Significant improvement of speech perception in noise - Significant improvement of hearing - Decreased disturbance caused by tinnitus; improvement was stable over time.	- CI use improved hearing and speech perception, and decreased tinnitus disturbance

Abbreviations: CI (cochlear implant), AHL (asymmetrical hearing loss), SSD (single-sided deafness), UHL (unilateral hearing loss), SSQ (speech, spatial and qualities of hearing scale), HSM (Hochmair–Schulz–Moser sentence test), CROS (contralateral routing of signal), BAHA (bone-anchored hearing aid), OLSA (Oldenburg sentence test), IOI-HA (international outcome inventory for hearing aids), HUI3 (health utilities index mark 3), VAS (visual analogue scale), THI (tinnitus handicap inventory), QoL (quality of life), APHAB (abbreviated profile of hearing aid benefit), FST (Freiburger numbers and monosyllabic test), TRQ (tinnitus reaction questionnaire), BKB-SIN (Bamford–Kowal–Bench sentence-in-noise), HINT (hearing in noise test), SRTs (speech reception thresholds), GBI (Glasgow benefit inventory), HADS (hospital anxiety depression scale), TTO (time trade off), HSM (Hochmair–Schulz–Moser sentences test), TQ (tinnitus questionnaire), SF-36 (*36*-Item *Short Form* Survey), AzBio test (Arizona biomedical institute sentence test), QoH (quality of hearing), NCIQ (Nijmegen cochlear implant questionnaire), PSQ (perceived stress questionnaire), COPE (Brief-COPE questionnaire), GAD-7 (generalized anxiety disorder questionnaire), OI (Oldenburg inventory), HRQoL (health-related quality of life), GFP (Gold field power), SHQ (sound hypersensitivity questionnaire), CAEPs (Cortical auditory evoked potentials), EQ-5D (European quality of life-five dimension), THQ (tinnitus handicap questionnaire), LIST (Leuven intelligibility sentence test), USTARR (Utrecht-sentence test with adaptive randomized roving levels), HINT (hearing in noise test), IEEE (Institute of Electrical and Electronics Engineers sentence test).

#### 3.3.3. Effect of Cochlear Implant on Other Factors

Along with tinnitus suppression, other criteria were assessed including speech comprehension in quiet and in noise, spatial hearing, hearing quality, speech perception and localization, sound quality, hyperacusis, work performance, psychological comorbidities, and QoL. Most studies reported improvement of sound localization and speech perception. The improvement of speech perception remained inconstant and oscillated during the first 6 months after implantation [81]. Speech recognition threshold (SRT) was improved [57]. No deterioration of speech performance was noted in the better hearing side with electric and acoustic signals integration [71]. Communication leading to less fatigue after a long workday and better work performance was also reported [80]. In addition, hyperacusis, evaluated using sound hypersensitivity questionnaire (SHQ) [55,66], as well as sound intolerance [78] were decreased among patients with CI. Furthermore, intracochlear electric stimulation improved QoL indexes and psychological comorbidities [44,72,75,80,83]. Taken together, these findings suggest that CI reduced tinnitus, restored hearing aspects, and improved QoL in SSD patients (Table 3 and Table 4).

## 4. Discussion

The present systematic review describes the effect of cochlear implantation on tinnitus in patients with SSD and disabling tinnitus. Reduction of tinnitus was reported in a relatively high number of studies (31 studies, 479 patients). No aggravation of tinnitus was reported in any patient. When compared to no treatment, CI was associated with better tinnitus suppression scores. These findings are encouraging in considering CI for SSD patients with disabling tinnitus, more specifically when conventional treatments fail to relieve the tinnitus. Although results are promising so far, the indication of CI for these patients is not yet widespread.

Most studies included in the present review assessed tinnitus using subjective tools; these are available in different languages, are not time consuming, and provide validated scores. VAS and THI were the most frequently used, followed by TQ. Although it could seem advantageous to not to be limited to a single tool, particularly since not all tools are validated for all languages, and some are more difficult to use than others, the heterogeneity of tools employed hampers comparison between studies. It is of note that objective tools were less likely to be used, which is possibly related to the difficulty of access to equipment required for electrophysiological and radiological assessments but also to the lack of available personnel with the skills to perform the assessments and interpret the results. These tools are, however, interesting in further understanding the mechanism of tinnitus reduction as well as the anatomical areas intervening in this process. It may also be helpful in identifying parameters that can predict prognosis. More generally, further research is needed to objectively assess treatment related physiological processes.

All SSD patients included in this review had disabling tinnitus, but the characteristics of their deafness were variable in terms of interval between onset and cochlear implantation, aetiology, and type of CI device. This makes it difficult to compare studies, but suggests treatment is successful independent of these factors. The risk of bias assessment showed a lack of precise inclusion criteria as well as a definition of the intervention in many studies. This emphasizes the need for a randomized clinical trial with clearly defined inclusion criteria and standard and clear intervention and rehabilitation protocols.

Whether tinnitus was evaluated as a primary complaint or not, CI succeeded to alleviate tinnitus. Studies in which tinnitus was evaluated as a primary complaint discussed several tinnitus characteristics including residual inhibition and recurrence of tinnitus after deactivation of implant. These studies were less likely to discuss hearing aspects or psychosocial benefits compared to studies where tinnitus was not the primary complaint.

Our review included all studies until December 2021. The present systematic review differs from previous published reviews in several ways. First, and to the best of our knowledge, this is the only review in which the listed studies have been divided into two groups depending on whether or not tinnitus was the primary complaint. The latter division permits reducing the risk of false-positive and false-negative errors. Second, the present review is not limited by the type of questionnaires used to assess tinnitus [50]: all validated multi-item questionnaires have been considered. In addition, studies using subjective assessment tools and studies using objective assessment tools were included. Audiological and neurophysiological levels of evidence were simultaneously considered when available. Last but not least, data on short- and long- term tinnitus suppression were analysed. The improvement of tinnitus, reflected by a significant reduction in various validated multi-item questionnaire scores, should strengthen considering CI in SSD with disabling tinnitus when conventional treatments are insufficient.

The present study has certain limitations; similar to the previously published systematic reviews, studies were mostly observational, and there was wide heterogeneity of tools used and a small sample size. This may preclude generalization of the results to a wider more heterogeneous population. Further studies with larger samples are needed to develop prediction models of tinnitus outcomes after cochlear implantation, where objective methods of tinnitus could be of interest.

## 5. Conclusions

In conclusion, this review included a large number of studies reporting the effectiveness of CI in suppressing disabling tinnitus in SSD patients when conventional treatment is insufficient. Tinnitus improvement is maintained in the long-term (>12 months). Considering the positive effect observed in all the studies, CI indication deserves to be more widely considered in such patients.

## Figures and Tables

**Figure 1 jcm-11-05664-f001:**
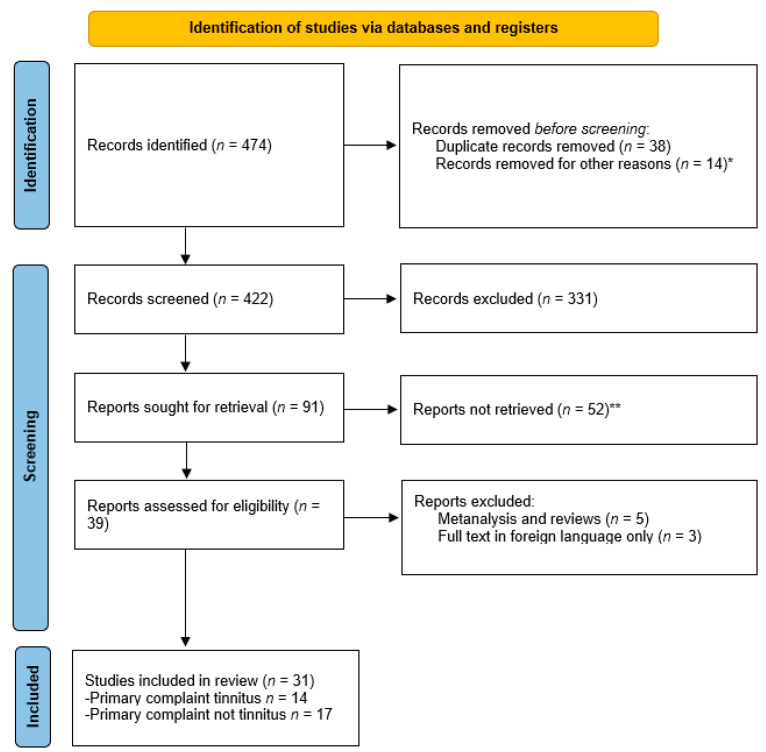
PRISMA 2020 flow chart diagram for updated systematic reviews which included searches of databases and study selection. Last date of search is December 2021. From: Page M J, et al. [52]. ** not relevant to the topic, ** Full text not found*.

**Table 3 jcm-11-05664-t003:** Characteristics of studies investigating CI in SSD patients with disabling tinnitus in which tinnitus was the primary complaint.

Study	Patients’ Criteria	n	Evaluation	Interval Studied	Results	Conclusion
Ahmed et al. [63]	CI in SSD and disabling tinnitus	13	*Questionnaires* - THI - TRS	3 months	- Significant improvement of THI and TRS	- CI is a treatment option of tinnitus suppression
Arts et al. [62]	CI in SSD and tinnitus (Intracochlear electrical stimulation vs. standard clinical CI).	10	*Tests* - VAS (tinnitus pitch and loudness matching) - RI *Questionnaires* - THI - TQ - HUI3 (HRQoL) - BDI (depression)	3 months	- Significant reduction of all tinnitus-related outcomes - Residual inhibition of tinnitus ranged from a few seconds to more than 30 min in 10 patients	- Significant reduction of tinnitus - No significant difference between intracochlear electrical stimulation and standard clinical CI on tinnitus outcomes - No significant difference between intracochlear electrical stimulation and standard clinical CI on QoL and depression outcomes
Holder et al. [64]	CI in SSD and tinnitus	12	*Tests* - CNC (word recognition). *Questionnaires* - THI	12 months	- Significant reduction of THI - Significant improvement of word recognition	- CI being an effective treatment option for SSD patients and tinnitus
Kleinjung et al. [60]	CI in SSD and severe tinnitus refractory to treatment	1	*Questionnaires* - VAS (tinnitus loudness and annoyance) - THI - TQ	3 months	- Distinct decrease in VAS, THI, and TQ - When CI is deactivated, tinnitus reoccurred only after presentation to loud noise	- Tinnitus completely disappeared 3 months after CI activation
Macias et al. [66]	CI in SDD and severe tinnitus	16	*Questionnaires* - VAS (tinnitus loudness) - THI - THS (hyperacusis) - HUI3 (QoL) - SSQ (Hearing)	12 months	- Significant decrease of VAS and THI - Significant decrease of hyperacusis handicap - Significant improvement of QoL and hearing - Residual inhibition of tinnitus was short-lasting with a median of less than 1 min	- Patients with SSD and concomitant severe tinnitus handicap were successfully treated with a CI
Mertens et al. [55]	CI in SSD and disabling tinnitus	23	*Questionnaires* - VAS (tinnitus loudness) - TQ - HQ (hyperacusis)	36 months	- Significant reduction of VAS and TQ - Significant difference of HQ scores - Residual inhibition of tinnitus is less than 1 min	- Tinnitus reduction remain stable up to 36 months
Mertens et al. [56]	CI in SSD and disabling tinnitus	11	*Questionnaires* - VAS (tinnitus loudness) - TQ	3 months	- Significant decrease of VAS and TQ	- CI can significantly reduce ipsilateral severe tinnitus in a subject with SSD.
Poncet-Wallet et al. [65]	CI in SSD and disabling tinnitus	26	*Tests* - Speech perception *Questionnaires* - VAS (tinnitus loudness and annoyance) - THI - TRQ - STSS	13 months	- Significant decrease of THI, TRQ, STSS, and VAS - Improvement of speech perception	- After 1 year of standard CI stimulation, 92% of patients reported a significant improvement in tinnitus
Punte et al. [59]	CI in SSD and severe tinnitus	26	*Tests* - TA (type, frequency, and loudness) *Questionnaires* - VAS (tinnitus loudness) - TQ (tinnitus distress)	6 months	- Significant reduction of VAS and TQ - When CI is deactivated, tinnitus reoccurred in 24 patients - Complete residual inhibition of tinnitus persists for at least 24 h (*n* = 2)	- Tinnitus loudness reduction remained stable over time - No difference on tinnitus reduction were observed according to tinnitus type - Tinnitus was completely abolished with CI activation in 3 patients
Punte et al. [68]	CI in SSD and severe tinnitus	7	*Tests* - TA (type, frequency, and loudness). *Questionnaires* - VAS (tinnitus loudness) - Psychoacoustic tinnitus loudness - TQ	6 months	- Significant decrease of VAS and TQ, and psychoacoustic tinnitus loudness after complete CI activation - When deactivated, tinnitus relapses and reoccurs to its original loudness in 6 patients - Complete residual inhibition of tinnitus persists for at least 24 h (*n* = 1)	- Tinnitus was completely abolished with CI activation in 1 patient - Limited reduction of VAS in 2 patients but coping with tinnitus is easier
Ramos et al. [61]	CI in SSD and disabling tinnitus refractory to prior treatment	6	*Tests* - TA (timbre, intensity, and minimum masking level) - HST (quantifying hyperacusis) - Hearing assessment *Questionnaires* - VAS - THI (perception and disability)	3 months	- Significant decrease or suppression of tinnitus perception and disability - Reduction of VAS - When CI is deactivated, improvement of tinnitus perception remained	- CI can reduce or suppress disabling tinnitus in patients with SSD
Song et al. [58]	CI in SSD and intractable tinnitus	9	*Tests* - EEG recording - sLORETA *Questionnaires* - NRS (tinnitus loudness) - TQ (subjective distress).	6 months	- Improvement in NRS and TQ	- Increased activities of AC and PCC, and increased functional connectivity between AC and PCC may be an unfavourable prognostic indicator after CI in patients with SSD
Van de Heyning et al. [54]	CI in SSD and severe intractable tinnitus unresponsive to treatment	22	*Questionnaires* - VAS (tinnitus loudness) - TQ (tinnitus distress)	24 months	- Significant reduction of VAS and TQ - When CI is deactivated, tinnitus reoccurred in 19 patients - Complete residual inhibition of tinnitus persists for at least 12 h (*n* = 3)	- Significant reduction in tinnitus when CI activated.
Zeng et al. [67]	CI in SSD and debilitating tinnitus refractory to treatment	1	*Tests* - Evoked and spontaneous cortical activities *Questionnaires* - VAS (tinnitus loudness)	720 s	- Low-rate low-level stimulus produced total tinnitus suppression - When stimulus is terminated, rebound in tinnitus was louder than baseline - Reduction of VAS	- Totally abolished tinnitus and restored normal brain activities

Abbreviations: AC (auditory cortex), BDI (Beck depression inventory), CI (cochlear implant), CNC (consonant-nucleus-consonant test), HQ (hyperacusis questionnaire), HST (hyperacusis test), HUI3 (health utilities index mark 3), NRS (numeric rating scale), PCC (posterior cingulate cortex), RI (residual inhibition), SHQ (sound hypersensitivity questionnaire), sLORETA (standardized low-resolution brain electromagnetic tomography), SSD (single-sided deafness), STSS (subjective tinnitus severity scale), TA (tinnitus analysis), THI (tinnitus handicap inventory), THS (test de Hipersensibilidad al sonido), TQ (tinnitus questionnaire), TRQ (tinnitus reaction questionnaire), TRS (tinnitus rating score), UHL (unilateral hearing loss), VAS (visual analogue scale).

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
