# Peer review of "Short- and Long-Term Effect of Cochlear Implantation on Disabling Tinnitus in Single-Sided Deafness Patients: A Systematic Review"

_jcm, 2022, doi:10.3390/jcm11195664_

Round 1

Reviewer 1 Report

Manuscript title: Evidence for cochlear implant indication in single-sided deafness with disabling tinnitus: a systematic review

Summary: It is a systematic review carried out to investigate the evidence for effectiveness of cochlear implant for disabling tinnitus. The review has included all the studies which have investigated the effect of cochlear implant on tinnitus irrespective of whether it was measured as a primary outcome or not. Studies which have used both subjective and objective measures of tinnitus are included in this review. The authors conclude that cochlear implant is effective in alleviating the tinnitus in individuals with single sided deafness.

            The manuscript covers the relevant references in the subject area. Most part of the manuscript is well written. Introduction can be expanded to clearly identify the knowledge gap. Need for the current review in the context of existing reviews is not clear. Furthermore, method and results section also needs more information. Results section can be elaborated and re-organized.

  General questions to help guide your review report for review articles:

  • Is the review clear, comprehensive and of relevance to the field? Is a gap in knowledge identified? – Review is clear. However, it needs improvisation in method and results section.
  • Was a similar review published recently and, if yes, is this current review still relevant and of interest to the scientific community? – Yes. The current review is still relevant and of interest to the scientific community. However, it is required to improvise the manuscript to highlight the additional information that this review is providing.
  • Are the cited references mostly recent publications (within the last 5 years) and relevant? Yes
  • Are any relevant citations omitted? Does it include an excessive number of self-citations? No
  • Are the statements and conclusions drawn coherent and supported by the listed citations? Yes
  • Are the figures/tables/images/schemes appropriate? Do they properly show the data? Are they easy to interpret and understand? Yes

During the manuscript evaluation, please rate the following aspects:

  • Interest to the Readers: Are the conclusions interesting for the readership of the journal? Will the paper attract a wide readership, or be of interest only to a limited number of people? (Please see the Aims and Scope of the journal.) – The manuscript will attract wide readership including professionals from various domains who are associated with cochlear implantation.
  • Overall Merit: Is there an overall benefit to publishing this work? Does the work advance the current knowledge? Do the authors address an important long-standing question with smart experiments? Do the authors present a negative result of a valid scientific hypothesis? Yes
  • English Level: Is the English language appropriate and understandable? Yes.

Specific comments

Abstract

Line 27. How is disabling tinnitus defined? That can be added here.

Line 29. The word “suppress” seems inappropriate. I suggest replacing it by any other appropriate term e.g., alleviate.

Line 29 and 31. “Conventional treatment” – what is included in conventional treatment? Can authors give an example here? Later in the manuscript, authors should provide the details as to what was considered as conventional treatment.

Line 35. Since “intracochlear electrical stimulation” is a separate technique to be used with CI, using this term seems inappropriate. Instead, “cochlear implant” can be used. (Also line 367)

Line 35 and 36. If the abstract length permits, I would suggest adding an example each for subjective and objective tools.

Line 37. Is “and” more appropriate than “but”?

Lines 39-41. Since investigating these parameters were not the objective of the study, the authors can append this. Instead, can add how the results were similar or different in studies that considered tinnitus as the primary outcome, highlight on the duration of long-term suppression etc.

Lines 41-44. Conclusion can be made more concise.

Introduction

Lines 50, 52, 54. Authors have used the terms “disabling tinnitus”, “handicap”, “disabilities”, “bothersome tinnitus” “incapacitating tinnitus” at different places throughout the manuscript. Authors should define it and make consistent usage of the same.

Line 53. This is more so for speech perception in noise. Can modify.

Line 56. “it can even also…” to modify.

Line 65. Why the words “but also” in this context?

Line 77. “Failure” does not seem to be correct term.

Lines 78-80. Authors should elaborate on how their manuscript it different from these existing reviews and what is the need for their review (some of this information has been included in the discussion section). Why is it important to consider studies that included tinnitus as primary outcome or not in different groups?

At the end, clear research question should be formulated.

Search strategy

Line 92. How “AND/OR” will work together?

Lines 104-105. Can mention the reviewer’s name.

Inclusion and exclusion criteria according to PICOS strategy should be clearly stated. Whether studies that assessed tinnitus pre- and post-operatively only were included or otherwise also? Which subjective and objective tools were included? What was the short- and long-term duration included? What age of the participants were considered? Whether participants had tried other treatment options for tinnitus previously and such other details.

Results

Figure 1. What is meant by “automation tools” and “other reasons” to remove the studies before screening? What are the reasons why studies were not “retrieved”?

Lines 146-148. Inclusion criteria with respect to hearing status of the contralateral ear and previous treatment experience appears to have combined. It can be separated.

How the lines 144-148 and lines 148-149 are different?

Lines 145-161. This paragraph is too big, and many results are put together. This can be reorganized.

Line 154-156. The sentence is not comprehensible.

Line 158. What is “standard fitting”? and what is absence of it?

Table 1. What is meant by deviation from intended intervention? Are the authors intending to mention about the additional interventions which was provided along with the CI? This needs clear mention in the manuscript as well (please refer to the previous comment). This applied to Table 2 as well.

The results section can have separate headings to report the studies which had tinnitus as the primary outcome and those which did not.

Line 228-229. Rephrase. What was the minimum duration?

Table 4. For most of the studies it is mentioned “improved tinnitus”. What does improved tinnitus mean?

Line 37-38 and Lines 274-276 seems to be contradicting, or difficult to understand. Also, what duration is considered as long-term?

Lines 283-285. Rephrase.

Line 294-295. Looks like an incomplete sentence.

Line 304. “Sound perception” same as “speech perception”? What is meant by “unstable”?

Discussion

Line 320. ‘s’ is missing in ‘tinnitus’

Lines 337-338 these study characteristics can also be included in results section either in text or in table (if available)

Lines 344-349. This paragraph should be rephrased, contains long sentences which are difficult to understand, and different concepts combined together.

Lines 350-359. Most of these points should be included and elaborated in the introduction section to strengthen the need for the review. Here, in discussion, it can be highlighted what are the additional information this review is providing in addition to the existing review. What are the contributions of the current review?

Line 367-368. What is refractory tinnitus?

Reviewer 2 Report

I do agree that review for SSD and cochlear implantation and tinnitus is necessary. But to meet the readers' expectation authors have to point out different aspect of these articles. I have two major concerns + recommendations. 

1. Scientific soundness for clinical research is provided with the design of the study. In this review, authors did not specifically searched for randomized controlled study (RCT). I do recommend the authors to analyze the RCTs excluding other designed studies. 

2. With the cochlear implant, tinnitus is expected to be relieved when the implant is turned on. What is in question is the other circumstances, when we turn of the cochlear implant. I think authors should find out when these studies measured the tinnitus, categorize the results into switch on or off. 

Round 2

Reviewer 2 Report

I have no further comment.